# Sex Difference Leads to Differential Gene Expression Patterns and Therapeutic Efficacy in Mucopolysaccharidosis IVA Murine Model Receiving AAV8 Gene Therapy

**DOI:** 10.3390/ijms232012693

**Published:** 2022-10-21

**Authors:** Matthew Piechnik, Paige C. Amendum, Kazuki Sawamoto, Molly Stapleton, Shaukat Khan, Nidhi Fnu, Victor Álvarez, Angelica Maria Herreño Pachon, Olivier Danos, Joseph T. Bruder, Subha Karumuthil-Melethil, Shunji Tomatsu

**Affiliations:** 1Nemours/Alfred I. DuPont Hospital for Children, Wilmington, DE 19803, USA; 2Sidney Kimmel Medical College, Thomas Jefferson University, Philadelphia, PA 19107, USA; 3Philadelphia College of Osteopathic Medicine, Philadelphia, PA 19131, USA; 4REGENXBIO Inc., Rockville, MD 20850, USA; 5Department of Pediatrics, Shimane University, Izumo 693-8501, Shimane, Japan

**Keywords:** adeno-associated virus, mucopolysaccharidoses, immune response

## Abstract

Adeno-associated virus (AAV) vector-based therapies can effectively correct some disease pathology in murine models with mucopolysaccharidoses. However, immunogenicity can limit therapeutic effect as immune responses target capsid proteins, transduced cells, and gene therapy products, ultimately resulting in loss of enzyme activity. Inherent differences in male versus female immune response can significantly impact AAV gene transfer. We aim to investigate sex differences in the immune response to AAV gene therapies in mice with mucopolysaccharidosis IVA (MPS IVA). MPS IVA mice, treated with different AAV vectors expressing human N-acetylgalactosamine 6-sulfate sulfatase (GALNS), demonstrated a more robust antibody response in female mice resulting in subsequent decreased GALNS enzyme activity and less therapeutic efficacy in tissue pathology relative to male mice. Under thyroxine-binding globulin promoter, neutralizing antibody titers in female mice were approximately 4.6-fold higher than in male mice, with GALNS enzyme activity levels approximately 6.8-fold lower. Overall, male mice treated with AAV-based gene therapy showed pathological improvement in the femur and tibial growth plates, ligaments, and articular cartilage as determined by contrasting differences in pathology scores compared to females. Cardiac histology revealed a failure to normalize vacuolation in females, in contrast, to complete correction in male mice. These findings promote the need for further determination of sex-based differences in response to AAV-mediated gene therapy related to developing treatments for MPS IVA.

## 1. Introduction

Mucopolysaccharidosis IVA (MPS IVA), or Morquio A Syndrome, is an autosomal recessive metabolic lysosomal disorder caused by a deficiency of the lysosomal N-acetylgalactosamine-6-sulfate sulfatase (GALNS) enzyme, which leads to the accumulation of keratan sulfate (KS) and chondroitin-6-sulfate (C6S), mainly in bone and cartilage [1]. Over time, the accumulation of KS and C6S in bone and cartilage leads to progressive hallmark clinical manifestations: short neck and stature, waddling gate, tendency to fall, spinal cord compression, pectus carinatum, kyphoscoliosis, hip dysplasia, and genu valgum. In general, patients with MPS IVA are diagnosed before 3 years of age with the initial signs (kyphosis, prominent chest, and forehead, short stature); however, some abnormalities (beaking sign at the lumbar vertebral body) on skeletal x-ray images appear at birth [2]. Phenotypic manifestation of the disease exists on a broad spectrum, and severe patients exhibit significant bone dysplasia and difficulty with the activities of daily living [3]. The mean age of death in MPS IVA patients is in the third decade, with respiratory failure causing two-thirds of patient deaths [4]. Current treatment for MPS IVA includes enzyme replacement therapy (ERT) and hematopoietic stem cell transplantation (HSCT), both of which have a limited impact on the characteristic skeletal dysplasia [5,6]. ERT, approved for clinical use for MPS IVA by the Food and Drug Administration in 2014, requires weekly infusions at a high cost [6], demonstrates a short half-life [7], and has not shown a significant impact in correcting growth and skeletal pathology [8]. HSCT has demonstrated practical clinical application in improving respiratory function and joint mobility while reducing surgical interventions [9,10,11]. Even though HSCT shows some impact on the improvement of bone and cartilage pathology, the approach is associated with other limitations such as the risk of graft versus host disease (GVHD), challenges in donor matching, graft rejection, and the need for well-trained staff and suitably equipped clinical facility for successful transplantation [5]. Given the limitations of available treatments, the focus of current research primarily involves novel therapeutic options such as gene therapy to potentially achieve therapeutic enzyme activity levels and bioavailability.

Gene therapy for MPS is a promising approach, with around 20 active or recruiting clinical trials investigating primarily viral-mediated gene therapy approaches (www.clinicaltrials.gov accessed on 11 November 2021) for MPS diseases. Viral gene therapy utilizes the transduction mechanism of a modified virus, most commonly the adeno-associated virus (AAV), to introduce corrected forms of the defective genetic sequence. AAV vectors have great potential due to their low pathogenicity, inability to self-replicate without a secondary viral infection, primarily episomal viral genome, and sufficient genetic payload (~4.7 kb) [12,13,14,15]. In addition, there are several serotypes of AAV with different transduction efficacies in particular tissue types, allowing for specificity in organ targeting. AAV has been shown to efficiently transduce target cells, induce high enzyme activity levels in target tissues, and achieve sustainable long-term enzyme expression [12,13]. However, AAV is limited by the immune responses generated against the AAV capsid, the transduced cells, and the transgene product. Innate and adaptive immune responses present a pronged response that reduces transduction rates and transgene product expression leading to lower enzyme activity levels and therapeutic efficacy [16]. Several strategies have been employed to overcome the immune response, including liver targeting to induce immune tolerance [17,18] and the use of steroids to inhibit T cell responses against the capsid to provide durable transgene expression [18,19] being the most well-known methods in clinical and preclinical models [20,21]. Liver-directed vector design permits AAV to utilize the tolerogenic nature of the liver microenvironment to ameliorate the immune response mounted by the body. AAV8 is the serotype of choice in developing a liver-targeting approach, given its high transduction efficacy for hepatocytes compared to other AAV serotypes, with dose-dependent transduction rates in murine models ranging from 90–95% [22]. Additionally, incorporating liver-specific promoters can increase hepatocyte specificity and reduce off-target transgene production [22]. These methods, taken in combination, have shown the potential for AAV-mediated gene therapy in treating MPS IVA and other inherited metabolic disorders.

Previously, we developed an AAV8 vector carrying thyroxine-binding globulin (TBG), a liver-specific promoter, and human GALNS to treat MPS IVA mice. We demonstrated partial efficacy in improving skeletal and cardiovascular pathology [23]. Disparities between male and female mice were noted concerning enzyme activity, antibody production, and histopathology. Observed differences in AAV efficacy between sexes in mice have been documented in the literature. In liver-targeted AAV2 and AAV5 vector administration, Davidoff et al. reported 7-fold higher stably transduced liver cells in male mice compared to female mice. Moreover, they reported that androgens play a critical role in transduction efficacy, proposing that upregulation of host nuclear regulatory proteins due to androgen presence plays a vital role in the differences seen. Davidoff et al. state that these sex differences were not observed in lung, kidney, or spleen [24]. In a separate study, Maguire et al. assessed differences in sex in the nervous system with the administration of AAV9 containing fluorescent gene products and a ubiquitous CAG (cytomegalovirus enhancer element, chicken β-actin promoter, and rabbit β-globin splice acceptor) promoter in endothelial cells, neurons, and astrocytes. They reported significantly higher fluorescence in female mice than male mice in the brain, with an inversely increased fluorescence in the abdomen (likely in the liver) in male mice compared to females [25]. While both mechanisms remain unknown, these reports suggest a significant relationship between sex and vector transduction mechanism that deserves attention.

Similar to sex-specific responses to AAV vectors seen in mice, male and female immune responses in humans also differ significantly, which may play a substantial role in viral-mediated gene therapy [26]. In mammals, females have been shown to mount more robust immune responses, innate and adaptive, compared to males [26]. Notably, the gender-based differences in the immune response are generally evolutionarily conserved [27,28,29,30,31,32,33]. Thus, to some extent, animal models reflect the differences observed between men and women. Sex-based biological theories suggest that the discrepancies observed are likely due to conserving metabolic resources. Male mice utilize more resources for sexual fitness, and female mice dedicate more resources to survival [34]. This causes women to be less susceptible to certain infectious diseases, such as ebola, hepatitis B, and tuberculosis. However, while heightened immunity to pathogens leads to a lower prevalence of infectious diseases in women, it can also cause increased symptoms and severity. Women have higher severity of several diseases, including human immunodeficiency virus (HIV), influenza, malaria, and zika. Regarding viral response, both female rats and women elicit a more robust innate response by demonstrating more significant cytokine upregulation induced by TLR-MyD88 pathway activation as a primary response to AAV vectors and the production of antibodies in response to viral challenges [35]. Furthermore, this difference in viral response also leads to sex-based differences in vaccine responses and a much higher incidence rate of autoimmune disease in women than men [26]. Given the importance of the immune response as a therapeutic hurdle in MPS IVA treatment, it becomes essential to analyze the depth of these differences to modify treatment protocols to reflect sex-based differences better.

In this study, we have analyzed sex differences in our previous data reported for liver-directed AAV8 gene therapy for MPS IVA murine models [23]. Furthermore, we present novel data analyzing sex differences between liver-specific and ubiquitous promoter cohorts. This novel investigation of combinatorial factors involved in the complex interaction between sex and immune response influencing therapeutic efficacy spotlights the importance of accounting for sex in future experimental designs.

## 2. Results

### 2.1. Sex, Promoter, and Mouse Model Comparison

In the previous study [23], MPS IVA knock-out (MKC; Galns^-/-^) mice were treated with an AAV8 vector containing TBG promoter coupled with codon-optimized human GALNS (hGALNS) with or without an N-terminal acidic amino acid (aspartic acid) octapeptide (D8) to target the bone preferentially. This group consisted of seven male and seven female mice. Two of seven male mice were treated with codon-optimized TBG-D8-GALNS (5 mice with TBG-GALNS), while four female mice were treated with TBG-D8-GALNS (3 mice with TBG-GALNS). Since the sex differences in immune responses were unexpected, vectors were assigned randomly so that an unequal number of male and female mice received each vector. This did not affect the goal of analyzing immune response due to AAV8 gene therapy, as all mice received uniform treatment with an unknown expectation of immune response incurred in male and female mice. All the mice received a uniform vector dose of 5 × 10^13^ GC/kg body weight at 4 weeks of age. This group of mice, treated with 5 × 10^13^ GC/kg body weight of vector with a TBG promoter in the previous study, is referred to as Group One in the results and figures.

The second study further divided mice by adding a ubiquitous CAG promoter (cytomegalovirus enhancer element, chicken β-actin promoter, and rabbit β-globin splice acceptor) cohort and also evaluating a higher dose cohort for the AAV8-TBG vectors. These cohorts of mice are included as Group Two and Group Three, respectively, in this report. Group Two comprised of mice treated with 5 × 10^13^ GC/kg body weight of AAV8 vector with the CAG promoter, and Group Three consisted of mice receiving 2 × 10^14^ GC/kg body weight of AAV8 vector with the TBG promoter.

Group Two consisted of six female mice (two received D8 appended treatment) and four male mice (two received D8 appended treatment). Group Three consisted of four female mice (two received D8 appended treatment) and eight male mice (five received D8 appended treatment). Mice receiving D8-GALNS treatment showed no significant improvement in pathology compared to those receiving the wild-type (WT) hGALNS [23]. AAV vector and study design were comparable in the data collection of the two studies (Figure 1).

### 2.2. Plasma Enzyme Activity, KS, and Anti-GALNS Antibodies in Males versus Females

In comparing male and female mice from both studies, male mice demonstrated higher plasma enzyme activity over time (Figure 2). In Group One, during the initial 2 weeks post AAV gene transfer, hGALNS enzyme activity levels sharply rose to supraphysiological levels in both male and female mice. Female mice then had decreasing plasma enzyme activity levels, reaching no more prolonged supraphysiological levels at 6 weeks post-treatment. This trend continued for the next six weeks, while the plasma enzyme activity level remained at supraphysiological levels in male mice throughout the monitoring period of 12 weeks after gene therapy. Additionally, male mice demonstrated significantly higher hGALNS activity levels than female mice from 6 weeks to 12 weeks after receiving AAV gene therapy (Figure 2A). Mice in Group Two had an elevation of enzyme levels shortly after injection. Male mice sustained higher enzyme activity levels than females throughout the remaining weeks. Female mice reduced the enzyme expression in Groups One and Two (Figure 2). In Group Three, mice demonstrated similar enzyme activity behavior to Group One. Female mice experienced a sharp rise, followed by a decrease in enzyme expression after 8 weeks old. Male mice sustained higher expression levels throughout the study (Figure 2B).

Regarding plasma KS levels, male mice generally exhibited lower plasma KS values than female mice across all factors, including age, promoter, and strain-matched cohorts post-treatment (Figure 3). In Group One, treated mice demonstrated normalized KS values from 2 weeks post-treatment. The females demonstrated a general higher plasma KS concentration throughout the study, with levels differing significantly from males at 12 weeks old (male to female, 15.9 ng/mL to 18 ng/mL [*p* = 0.04]) and 14 weeks old (male to female, 14.8 to 18.3 ng/mL [*p* = 0.01]) (Figure 3A). In Group Two, male mice treated with the CAG promoter construct demonstrated normalized plasma KS values, whereas female mice showed KS levels above WT levels; however, both male and female mice significantly improved KS levels compared to the untreated cohort. Female mice had higher KS levels than male mice from 6 through 16 weeks old (Figure 3B). In Group Three, male mice had lower KS levels than female mice and achieved normalized KS values between 6 weeks (6 weeks, female to male, 44.4 ng/mL to 27.7 ng/mL; 12 weeks, female to male, 49.1 ng/mL to 30.8 ng/mL) and 16 weeks (female to male, 54.0 ng/mL to 37.7 ng/mL). Female mice had decreased but not normalized KS values post-treatment throughout the study (Figure 3C).

Anti-GALNS antibody values reflected the humoral aspect of the immune response generated against AAV-mediated gene therapy. In Group One, both males and females steadily increased antibody titers, with females averaging higher titers than their male counterparts (Figure 4A). Males and females in Group Two tended to exhibit similar antibody titers until 6 weeks, and female mice had a higher antibody with age and ramped up at 16 weeks old (Figure 4B). In Group Three, females demonstrated higher anti-GALNS antibodies from 6 weeks old and remained higher till 16 weeks old (Figure 4C).

### 2.3. Tissue Enzyme Activity and KS Concentration

Tissues were harvested at necropsy 12 weeks after AAV gene therapy injection, and enzyme activity levels and KS concentrations were determined. In Group One, enzyme activity was measured in the liver, heart, kidney, lung, spleen, and bone (Figure 5A). The same tissues were analyzed in Groups Two and Three (Figure 5B). In Group One, male mice generally demonstrated higher enzyme activity levels than females in all tissues, significantly higher in heart, kidney, lung, spleen, and bone tissue (Figure 5A). Furthermore, male and female mice achieved supraphysiological hGALNS activity levels in liver tissue (Figure 5A). In Group Two, male mice demonstrated significantly higher activity than females in heart, spleen, and bone (Figure 5B). Despite this difference, males and females achieved supraphysiological levels in the liver, heart, spleen, and bone (Figure 5B). In Group Three, mice displayed activity levels similar to Group One despite the 4-fold higher dose. Males achieved higher levels in the heart, spleen, lung, and bone (Figure 5B). Comparing Group One to Group Two, female mice treated with the CAG promoter construct demonstrated higher activity levels in kidney, lung, spleen, and bone tissue than female mice treated with AAV vector with TBG promoter construct. In contrast, male mice demonstrated similar activity levels in all tissues except bone, which had increased activity. A wide range of variability among individuals was observed in liver tissue across both sexes and vectors (Figure 5). Additionally, analyses of the correlation between the activity levels detected in plasma at 16 weeks and in tissues in groups treated with TBG, a liver specific promoter showed that females in Group One have a significant correlation in the heart while females in Group Three in spleen and lung. However, in males, there was no correlation between activity detected in plasma and tissues.

In Group One, tissue KS concentrations were measured in the liver and lung, while concentrations were only measured in the liver for Groups Two and Three. In Group One, female mice had higher KS levels than male mice in both lung and liver (Figure 6A). AAV gene transfer reduced the KS concentration in the liver of both males and females but were still higher than normal levels, whereas the lung KS levels were normalized in males. In Groups Two and Three, treated with the same doses, liver KS levels were not different between male and female mice (Figure 6B).

### 2.4. Bone and Heart Histopathology

In evaluating tibial and femoral growth plates, articular cartilage, ligament, and menisci, mice were assessed for vacuolation severity (all tissues) and columnar structure (growth plate) using a pathology scoring system [23] (see Materials and Methods). In Group One, male mice generally demonstrated a higher level of vacuolation and more disorganized columnar structure than female mice in the femoral and tibial growth plates (Figure 7A). In Group Two, both male and female mice demonstrated improvement in femoral growth plate pathology, whereas only males showed significant improvement in the tibial growth plate (Figure 7B). Compared to Group One, males treated with CAG promoter construct (Group Two) demonstrated more normalized tissue structure than females. This tendency was also observed in Group Three. Males significantly improved from the untreated group, while females did not (Figure 7C). Histopathological Analysis by toluidine blue staining showed that the vacuolation in the growth plate of the femoral and tibia bone was more profound in females treated with either the CAG or TBG promoter vector than in males (Figure 8A,B). In Groups Two and Three, male mice significantly improved more than females (Figure 7B,C).

In Group One, the femoral articular cartilage in both male and female mice exhibited score improvement compared to the untreated group, and in the tibial articular cartilage, only females showed an improvement, although not significant in both cases(Figure 7D). In Group Two, both male and female mice showed significant improvement in articular cartilage pathology in both the femur and the tibia compared to the untreated cohort (Figure 7E). In Group Three, female mice showed no improvement compared to the untreated group, while males demonstrated significant corrected pathology (Figure 7F).

Femoral and tibial ligament analysis revealed a similar trend in improving pathology scores. In Group One, male mice experienced higher pathology scores in the femoral ligament than female mice. Meniscus evaluation also displayed a stark contrast between males and females, with females presenting with significantly lower pathology scores than males, further demonstrating a prevalent dichotomy in the treatment efficacies (Figure 7G). Both males and females in Group Two exhibited similar levels of femoral ligament pathology severity to untreated mice. In contrast, the tibial ligament showed similar improvement in both males and females compared to the untreated group (Figure 7H). In Group Three, ligament pathology was less severe in males than in females. Both the femoral and tibial ligaments were partially rescued in the males, while females exhibited no significant difference from the untreated cohort (Figure 7I). Ligament vacuolation was worse in females than males across both vector groups. In addition, in the growth plate and articular cartilage, males demonstrated better outcomes when treated with CAG promoter than with TBG promoter vectors (Figure 7A–F).

Heart tissue analysis of mice was performed for Group One, and tissues from the base, valves, and myocardium were analyzed for severity rating. Males demonstrated basal, valvular, and myocardial tissue correction with complete normalization in all individuals. Female mice exhibited partial correction in all individuals for basal and valvular tissues and complete normalization in myocardial tissue compared to the untreated group (Figure 9).

## 3. Discussion

This study identified sex differences in plasma enzyme activity, blood KS concentration, and anti-GALNS antibody levels in MPS IVA knock-out (MKC; Galns^-/-^) mice treated with AAV8 gene therapy. Overall, female mice demonstrated less treatment efficacy as determined by higher levels of antibody production, lower levels of GALNS activity, and a greater concentration of KS in the blood than male mice. These results were maintained across promoter and/or dose design cohorts, suggesting that sex plays a critical role in determining therapeutic efficacy in this animal model. Moreover, the untreated cohort had no significant difference between male and female mice in KS levels and pathological scores. The interplay of increased enzyme activity, reduced storage substrate concentration, and decreased antibody formation improved histopathology in both male and female mice compared to the untreated cohorts, although with a more limited effect in females.

The mechanism of AAV transduction is multifaceted, with several critical steps, including endocytosis of the viral particle, endosomal escape, uncoating of the capsid, and second-strand synthesis from the single-stranded DNA vector genome [36,37,38,39]. Davidoff et al. demonstrated that the binding of host nuclear proteins in hepatocytes, crucial for the transcription of vector mRNA, is androgen-dependent and that this relationship is not observed in other non-hepatic tissues [24]. Our results fit this theory, with observable differences between the male and female cohorts. Notably, Dodge et al. completed similar work with an analysis of acid sphingomyelinase enzyme activity after AAV2/1 and AAV2/2 delivery in the brain between male and female mice, explicitly looking at variability in enzyme activity of females in different stages of the estrous cycle. Their findings suggest a negative correlation between circulating female hormone levels and enzyme activity [40]. While we injected the AAV vector into mice at the earlier end of the puberty spectrum, it remains possible that female hormones in circulation played a role in reducing enzyme activity over time.

The immune response disparity between male and female mice in this study was similar to the pattern of immunogenic sex differences previously demonstrated [24,25]. Both innate and adaptive immune responses are crucial in limiting successful AAV transduction. The most prominent innate immune response to AAV is mediated by the TLR9-MyD88 pathway, which initiates inflammatory cytokine production via the NF-κB pathway, and the adaptive immune response begins [35]. In evaluating sex-based differences, Torcia et al. reported that stimulation of TLR9 induced higher production of the immunosuppressive cytokine IL10 in males, which reduced the overall strength of the male innate immune response [41]. This compounded effect permits male mice to exhibit a weaker immune defense allowing a more significant opportunity for successful transduction.

The humoral responses to the vector capsid and transgene products are the most significant in limiting therapeutic efficacy and preventing long-term gene expression and recurrent treatments. In our study, female mice showed higher anti-GALNS antibody titers than males, consistent with the expected more robust humoral response in females. Likely, the differences observed between males and females link back to the interactions between sex-specific hormones and their propensity for immune-modulatory receptors such as the NF-κB, cJun, and interferon regulatory factor 1 pathways [42,43]. The immune response to anti-AAV antibodies were not measured in this study. The humoral immune response to AAV shows that anti-capsid antibodies raised impact AAV transduction inefficiency [44]. The clinical significance of the immune response is reported in the recent CHAMPIONS clinical trial of SB-913, an AAV/zinc finger nuclease (ZFN)-mediated gene therapy for MPS II. Diminishing efficacy was shown in plasma iduronate-2-sulfate (IDS) activity with a correlating increase in the liver enzyme ALT, due to a cytotoxic response against transduced liver cells [45]. Because of shortage of samples, we could not measure AST and ALT liver enzymes; however, it would be important to know whether there is some sex difference in AST and ALT levels.

The impact of sex hormone regulation on viral entry mechanism and the immune response manifests in the sex-dependent difference in the therapeutic efficacy of correcting the MPS IVA phenotype in our mouse model. Reduced vacuolation and organized columnar structure are critical in developing normal bone growth, and adequate correction early in life can ameliorate the severity of bone dysplasia experienced by MPS IVA patients. Although the differences between males and females were insignificant, the averages reflect better outcomes in males than females. Greater sample size may further clarify any differences. However, bone pathology correction is a multifactorial problem arising from the poor vascularization of cartilaginous tissue.

Based on the plasma hGALNS activity and anti-GALNS antibody levels in female mice, we demonstrated declining hGALNS activity over time which directly correlated with increasing anti-GALNS antibody titers (Figure 2, Figure 3 and Figure 4). Male mice provided better outcomes with lower antibodies and higher stable enzyme activities during the treatment period than female mice despite the difference in the promoter or dose. Furthermore, male mice had higher enzyme activities in heart, lung, spleen, and bone. Male mice in Groups Two and Three demonstrated lower pathology scores for tibial and femoral growth plates and articular cartilage. On the other hand, the activity reported in tissues in groups 1 and 3 treated with TBG promoter, specifically expressed in hepatocytes, is due to the re-captured from circulating enzyme as the results of the correlation shown.

There were some discrepancies in our data, particularly in the pathology scoring of Groups One and Three, while both groups received the TBG promoter (Figure 7). There is an increase in hGALNS activity in bone when comparing Group Three to Group One (Figure 5), and males have higher activity in both groups. Female mice in Group One had lower pathology scores than male mice, and female mice in Group Three had higher pathology scores than male mice in tibial and femoral growth plates and articular cartilage. These differences are likely related to the difference in doses between the two groups or could arise due to small sample sizes or variations in disease presentation or progression in individual mice. The exact mechanism of pathology improvement remains unknown, requiring further large cohort study.

Identifying the differences in male and female interactions with AAV-mediated gene therapy is important because many preclinical studies do not consider this intrinsic factor in their study design. In 2015, the United States Government Accountability Office (USGAO) reported that the National Institutes of Health (NIH) did not make readily available data on women enrolled in human clinical trials and did not ensure that women were adequately represented in clinical trials. By failing to account for the variable of biological sex, these studies may have misrepresented the corresponding effects of sex [46]. Incorporating this issue begins at the preclinical level, where adequate reporting and/or selection of sex and evaluating sex differences is essential for discerning accurate results. Many preclinical studies do not report sex or control for sex-based effects [47,48,49]. Therefore, the inclusion of sex as a critical variable in developing sound scientific processes will permit improved therapeutic efficacy for all patients, a necessity demonstrated by our results in this study.

## 4. Materials and Methods

Materials and Methods for the first study data followed the methods described in Sawamoto et al. [23]. Similar or identical materials and methods were used to develop the second study experiments.

### 4.1. AAV Vector and Cassette Design

AAV cassettes were designed with either D8-hGALNS or native hGALNS downstream of a TBG promoter or a ubiquitous CAG promoter, including a cytomegalovirus enhancer element, a chicken β-actin promoter, and an intron incorporated into a single promoter sequence (Figure 1). All the hGALNS sequences were codon-optimized. The D8-hGALNS region contained codon-optimized hGALNS preceded by a bone-targeting aspartic acid octapeptide (GACGACGATGATGACGATGACGAC). The hGALNS sequences (GenScript, Piscataway, NJ) were incorporated into the vector upstream of a rabbit β-globin polyadenylation tail. Our previous study demonstrated successful in vitro enzymatic activity in Huh7 cells [23].

Proprietary protocols developed at REGENXBIO (REGENXBIO Inc. Rockville, MD, USA) were followed to produce all research-grade AAV vectors used in the studies described here. Briefly, triple transfection of HEK293 cells was performed with the AAV8 capsid plasmid, helper plasmid, and the respective transgene plasmid. Affinity chromatography was performed on cell culture supernatant, and the purified vectors were titered utilizing Digital Droplet PCR (Biorad, Hercules, CA, USA).

### 4.2. Murine Models

As previously described, the MPS IVA knock-out mouse (MKC; Galns^-/-^) was developed via targeted disruption of exon 2 of GALNS [23,50]. All the studies described in this manuscript used this mouse model. Like MPS IVA patients, this mouse model demonstrated minimal to no hGALNS enzyme activity in circulation, higher blood KS concentration, and vacuolation in various cells due to glycosaminoglycan accumulation. At two weeks old, mice were genotyped via PCR and selected for homozygous Galns^-/-^. In the first study (Group One), male and female mice were administered 5 × 10^13^ GC/kg of either AAV8 TBG-D8-hGALNS or AAV8 TBG-hGALNS via tail-vein injection at 4 weeks of age. The second study separated male and female mice into CAG or TBG promoter cohorts. The mice treated with either the AAV8-CAG-D8-hGALNS or the AAV8-CAG-hGALNS vector received 5 × 10^13^ GC/kg via tail-vein injection at 4 weeks of age (Group Two). The mice treated with the AAV8-TBG-D8-hGALNS or the AAV8-TBG-hGALNS vector received 2 × 10^14^ GC/kg via tail-vein injection at 4 weeks old (Group Three). The untreated and WT mice in both studies received phosphate-buffered saline (PBS) as a control. All injections were approximately 100 µL in total dose volume. The Institutional Animal Care and Use Committee of Nemours/Alfred I. duPont Hospital for Children approved our animal care and experimentation.

### 4.3. Blood and Tissue Collection

Following the blood and tissue collection timeline (Figure 1), approximately 100 µL of blood was collected from all cohorts in EDTA tubes (Becton Dickinson, Franklin Lakes, NJ, USA). Blood tubes were then centrifuged at 8000 rpm for 10 min, at which point plasma was extracted and stored at −20 °C for GALNS activity assays and KS measurements. Tissue collection occurred at 16 weeks of age when mice were euthanized via CO_2_ chamber and perfused with 20 mL of 0.9% normal saline. Liver, kidney, lung, spleen, heart, trachea (in the first study only), femur, tibia, and knee joint were collected, snap-frozen, and stored at −80 °C until KS and GALNS activity assays. Some bone and heart tissues were collected and stored in 10% neutral buffered formalin and then transferred to 2% glutaraldehyde/4% paraformaldehyde solution for histopathological analysis.

### 4.4. Plasma and Tissue GALNS Activity Assay

The previously described protocols [51] determined activity levels of GALNS. Frozen tissues were homogenized with 25 mmol/L Tris-HCl (pH 7.2) and 1 mmol/L phenylmethylsulfonyl fluoride homogenization buffer. The tissue lysate or previously collected plasma was combined with 22 mM 4-methylumbelliferyl-β-galactopyranoside-6-sulfate (Research Products International, Mount Prospect, IL, USA) in 0.1 M NaCl/0.1 M sodium acetate (pH 4.3) in a 96-well plate and incubated at 37 °C for 16 h. 10 mg/mL β-galactosidase from Aspergillus oryzae (Sigma-Aldrich, St. Louis, MO, USA) in 0.1 M NaCl/0.1 M sodium acetate (pH 4.3) was added to the reaction mixture and incubated at 37 °C for an additional 1 h. 1 M glycine, NaOH (pH 10.5) was then added to arrest the reaction. Plates were transferred to a PerkinElmer Victor X4 plate reader (PerkinElmer, Waltham, MA, USA) and excited at 366 nm with an emission read at 450 nm. Results were recorded in nanomoles of 4-methylumbelliferone released per hour per microliter of plasma or milligram of protein. Conversion of plasma to protein concentration was determined by a bicinchoninic acid (BCA) protein assay kit (Thermo Fisher Scientific, Waltham, MA, USA).

### 4.5. Tissue GAG Extraction and Analysis

GAG extraction was based on the procedure outlined in Mochizuki et al. [51]. Tissue samples were frozen in liquid nitrogen and combined with acetone as a defatting solvent in a homogenizer. The homogenized mixture was centrifuged for 30 min at 4 °C, and the pellet was dried via centrifugation vacuum, suspended in 0.5 M NaOH, and incubated at 50 °C for 2 h, removing GAG chains from core proteins. The solution was neutralized with 1 M HCl, and NaCl was added to achieve a concentration of 3 M. The sample was then centrifuged to remove nucleotides, and the supernatant was collected. The pH of the supernatant was adjusted to below 1.0 with excess 1 M HCl. Centrifugation was then repeated to remove precipitated proteins, and the supernatant was then neutralized with excess 1 M NaOH. 2 volumes of 1.3% potassium acetate (prepared in ethanol) were added to precipitate GAGs. Tubes with crude GAGs were centrifuged at 4 °C for 30 min and washed with 80% ethanol. The GAGs were dried out and reconstituted in 50 mM Tris-HCl pH 7.0 and stored at −20 °C for analysis by LC-MS/MS.

Both plasma and tissue KS concentrations were measured via liquid chromatography-tandem mass spectrometry (LC-MS/MS) as outlined previously [52,53,54,55,56,57]. Either plasma or homogenized tissue lysate samples were combined with 50 mM Tris-HCl (pH 7.0) and added into a 96-well Omega 10 K MWCO (molecular weight cutoff) filter plate (Pall Corporation, Port Washington, NY) on a 96-well receiver plate. Centrifugation was performed at 2500 rpm for 15 min. The filter plate was transferred to a new receiver plate, and 50 mM Tris-HCl (pH 7.0), 5 mg/mL chondrosine as internal standard (IS), and 1 mU keratanase II were added. Overnight incubation was performed at 37 °C. Samples were then centrifuged at 2500 rpm for 15 min. The LC-MS/MS was performed using a 1290 Infinity LC system with a 6460 triple quad mass spectrometer (Agilent Technologies, Palo Alto, CA, USA). Disaccharide separation was performed on a Hypercarb column (2.0 mm inner diameter [i.d.], 50 mm long, 5-mm particles; Thermo Fisher Scientific, Waltham, MA, USA) at 60 °C. The mobile phase was developed as a gradient elution of 5 mM ammonium acetate (pH 11.0) (solution A) to 100% acetonitrile (solution B). The flow rate was established at 0.7 mL/min with a gradient of 0 min, 100% solution A; 1 min, 70% solution A; 2 min, 70% solution A; 2.20 min, 0% solution A; 2.60 min, 0% solution A; 2.61 min, 100% solution A; 5 min, 100% solution A. Electrospray ionization was set for the mass spectrometer in the negative ion mode (Agilent Jet Stream technology) with an injection volume of 5 μL and a sample run time of 5 min. Known m/z ratios for initial and product ions were used to determine and quantify each disaccharide (IS, 354.3/193.1; mono-sulfated KS, 462/97).

### 4.6. Anti-GALNS Antibody Assay

Quantification of anti-GALNS antibodies was determined via ELISA assay as described previously [58]. 96-well microtiter plate were coated overnight with 2 mg/mL purified rhGALNS (R&D Systems, Minneapolis, MN, USA) in 15 mM Na_2_CO_3_, 35 mM NaHCO_3_, and 0.02% NaN_3_ (pH 9.6). The plate was then triple-washed with Tris-buffered saline (TBS)-T (10 mM Tris [pH 7.5], 150 mM NaCl, 0.05% Tween 20), then blocked for 1 h at 20 °C with 3% bovine serum albumin in PBS (pH 7.2). Mouse plasma in TBS-T at 100 times dilution was added to the wells, and incubation was performed for 2.5 h at 37 °C. Wells were then rewashed four times in TBS-T. TBS-T with a 1:1000 dilution of peroxidase-conjugated goat anti-mouse immunoglobulin G (IgG) (ThermoFisher Scientific, Waltham, MA, USA) was added, and incubation was performed for 1 h at 20 °C. Samples were then triple-washed with TBS-T and then double-washed in TBS (10 mMTris [pH 7.5], 150 mM NaCl). Peroxidase substrate (ABTS solution, Invitrogen, Carlsbad, CA, USA) was added at 100 μL per well, and samples were then incubated for 30 min at 20 °C. 1% SDS was added to arrest the reaction. The plates were transferred to a PerkinElmer Victor X4 plate reader (PerkinElmer, Waltham, MA, USA) at 410 nm. In the first study, plates were reported as absorbance in optical density units. In the second study, the samples were run with known rabbit GALNS polyclonal IgG concentrations in serial dilution to develop standards. Linear regression was performed to convert optical density to μg of IgG per mL of plasma for cohort data.

### 4.7. Histopathological Analysis

Tissue staining and analysis were performed as described by Tomatsu et al. [59]. The knee joint, mitral heart valve, and myocardium were collected 12 weeks post-treatment to evaluate vacuolation severity via light microscopy. Tissue samples were fixed in 2% paraformaldehyde, 4% glutaraldehyde in PBS, then post-fixed in osmium tetroxide, and embedded in Spurr’s resin. Toluidine blue-stained 0.5-mm-thick sections were examined under light microscopy. Assessment of reduction of storage materials were described previously [60,61]. Briefly, tissues from treated, untreated, and wild-type mice were evaluated for reduction in storage without knowledge of their treatment. The analysis was performed with three blind readers to evaluate the severity of a scalar model of normal to severe (0 to 3) for the femoral and tibial growth plate, articular cartilage, and ligaments. “No storage or very slight” was 0, “slight but obvious” was 1, “moderate” was 2, and “marked” was 3. In the first study, cardiac tissue pathology was evaluated similarly on a normal to severity scale of 0 to 1. Averages of individual severity scores were calculated and reported. Mann–Whitney U test was used to compare the score between untreated and treated mice.

### 4.8. Statistical Analysis

Data were reported as means with standard deviations. Comparison analysis was performed with two-tailed *t*-tests assuming unequal variance. One-way ANOVA tests were performed with Bonferroni’s post hoc test using GraphPad Prism 5.0 (GraphPad, San Diego, CA, USA). Statistical significance of difference was determined to be *p*-values less than 0.05.

## Figures and Tables

**Figure 1 ijms-23-12693-f001:**
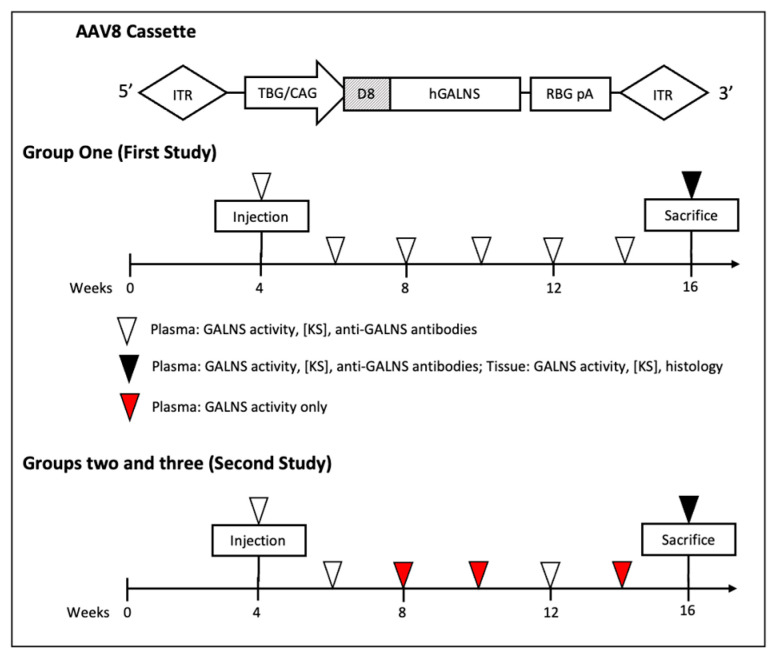
Cassette and study design. The experimental scheme of the first study was described by Sawamoto et al. [23]. AAV8: adeno-associated virus 8, ITR: inverted terminal repeat, TBG: thyroxine-binding globulin, CAG: a ubiquitous promoter including a cytomegalovirus enhancer element, a chicken β-actin promoter, and an intron, D8: aspartic acid octapeptide, hGALNS: human N-acetylgalactosamine-6-sulfate sulfatase, RBG pA: rabbit β globin poly-A tail.

**Figure 2 ijms-23-12693-f002:**
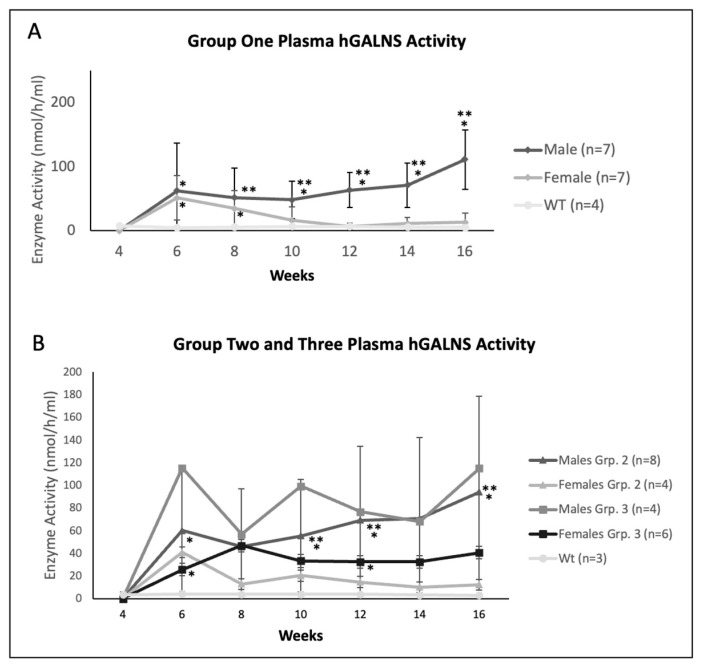
Plasma hGALNS activity level of (**A**) Group One (mice treated with the TBG promoter; 5 × 10^13^ GC/kg body weight at 4 weeks old), (**B**) Group Two (mice treated with CAG promoter; 5 × 10^13^ GC/kg body weight at 4 weeks old), and Group Three (mice treated with TBG promoter; 2 × 10^14^ GC/kg body weight at 4 weeks old). Error bars display standard deviation. * indicates a two-tailed *t*-test *p*-value of <0.05 compared to the WT cohort. ** indicates a two-tailed *t*-test *p*-value of <0.05 compared to the opposite sex cohort. hGALNS: human N-acetylgalactosamine-6-sulfate sulfatase, TBG: thyroxine-binding globulin, CAG: a ubiquitous promoter including a cytomegalovirus enhancer element, a chicken β-actin promoter, and an intron, WT: wild-type.

**Figure 3 ijms-23-12693-f003:**
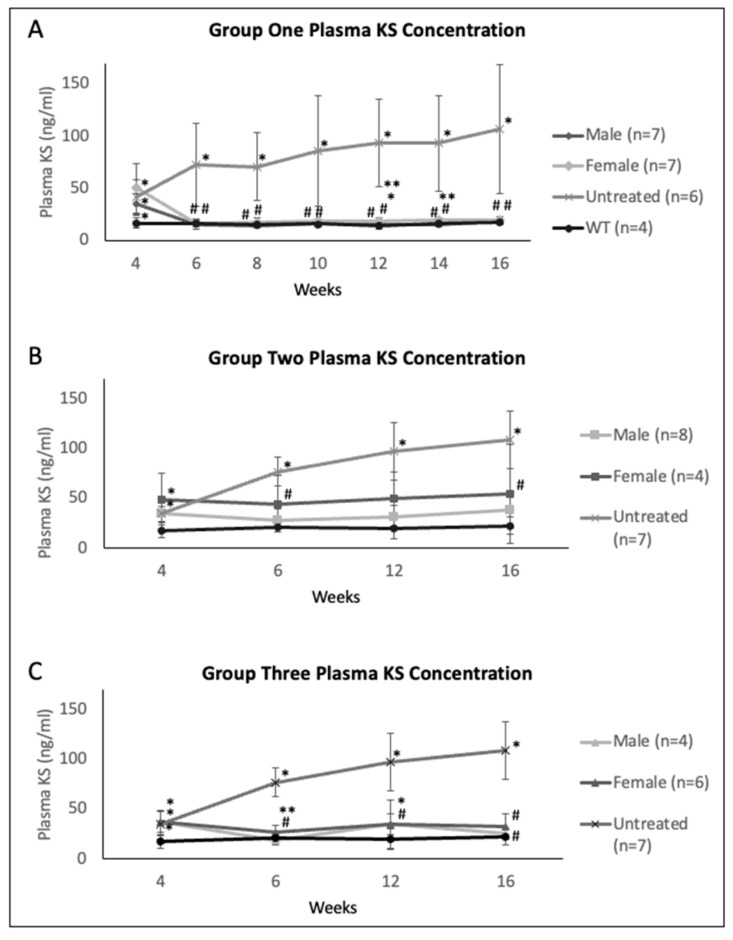
Plasma KS concentration of (**A**) Group One (mice treated with the TBG promoter; 5 × 10^13^ GC/kg body weight), (**B**) Group Two (mice treated with the CAG promoter; 5 × 10^13^ GC/kg body weight), and (**C**) Group Three (mice treated with the TBG promoter; 2 × 10^14^ GC/kg body weight). Error bars display standard deviation. * indicates a two-tailed *t*-test *p*-value of <0.05 compared to the WT cohort. ** indicates a two-tailed *t*-test *p*-value of <0.05 compared to the opposite sex. # indicates a two-tailed *t*-test *p*-value of <0.05 compared to the untreated cohort. KS: keratan sulfate, TBG: thyroxine-binding globulin, CAG: a ubiquitous promoter including a cytomegalovirus enhancer element, a chicken β-actin promoter, and an intron, WT: wild-type.

**Figure 4 ijms-23-12693-f004:**
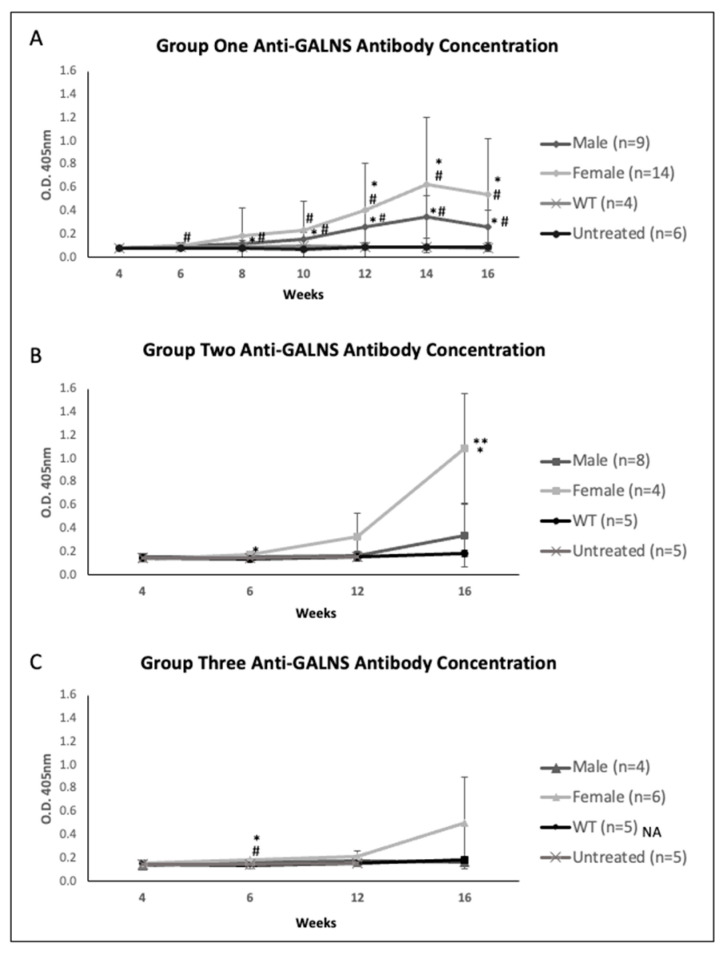
Plasma anti-GALNS antibody concentration of (**A**) Group One (mice treated with the TBG promoter; 5 × 10^13^ GC/kg body weight), (**B**) Group Two (mice treated with the CAG promoter; 5 × 10^13^ GC/kg body weight), and (**C**) Group Three (mice treated with the TBG promoter; 2 × 10^14^ GC/kg body weight). Concentrations are displayed as optical density absorption rates at the 405 nm wavelength. Error bars display standard deviation. * indicates a two-tailed *t*-test *p*-value of <0.05 compared to the WT cohort. ** indicates a two-tailed *t*-test *p*-value of <0.05 compared to the opposite sex. # indicates a two-tailed *t*-test *p*-value of <0.05 compared to the untreated cohort. GALNS: N-acetylgalactosamine-6-sulfate sulfatase, TBG: thyroxine-binding globulin, CAG: a ubiquitous promoter including a cytomegalovirus enhancer element, a chicken β-actin promoter, and an intron, WT: wild-type.

**Figure 5 ijms-23-12693-f005:**
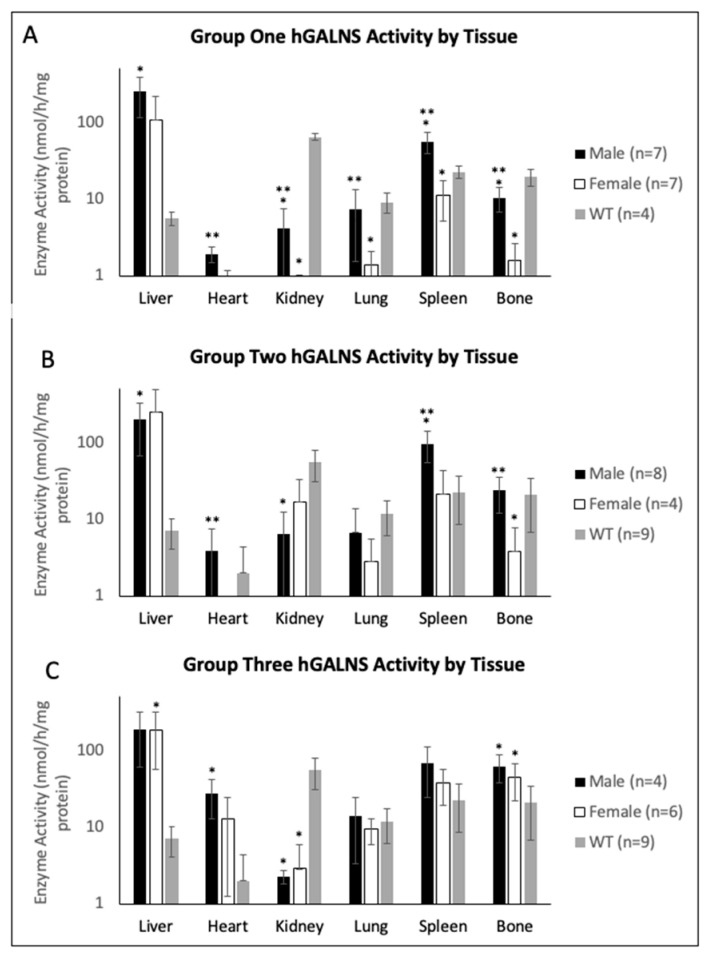
hGALNS activity in tissues for (**A**) Group One (mice treated with the TBG promoter; 5 × 10^13^ GC/kg body weight), (**B**) Group Two (mice treated with the CAG promoter; 5 × 10^13^ GC/kg body weight), and (**C**) Group Three (mice treated with the TBG promoter; 2 × 10^14^ GC/kg body weight). Error bars display standard deviation. * indicates a two-tailed *t*-test *p*-value of <0.05 compared to the WT cohort. ** indicates a two-tailed *t*-test *p*-value of <0.05 compared to the opposite sex. Graphs are plotted on a logarithmic scale to better display distributions. hGALNS: human N-acetylgalactosamine-6-sulfate sulfatase, TBG: thyroxine-binding globulin, CAG: a ubiquitous promoter including a cytomegalovirus enhancer element, a chicken β-actin promoter, and an intron.

**Figure 6 ijms-23-12693-f006:**
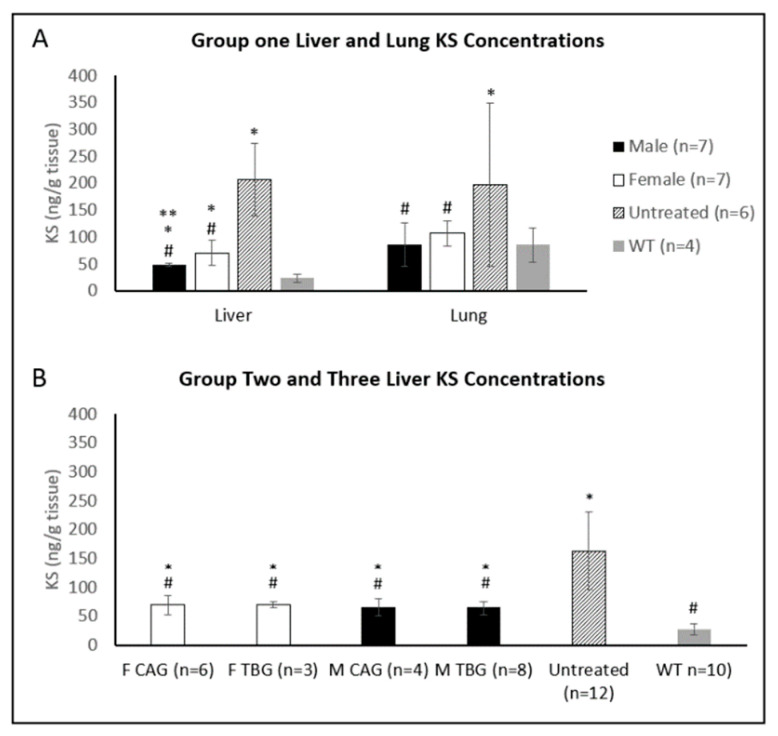
(**A**) KS concentrations in liver and lung tissue for Group One (mice treated with the TBG promoter) and (**B**) KS concentrations in liver for Groups Two and Three (mice treated with the CAG and TBG promoter, respectively). Error bars display standard deviation. * indicates a two-tailed *t*-test *p*-value of <0.05 compared to the WT cohort. ** indicates a two-tailed *t*-test *p*-value of <0.05 compared to the opposite sex. # indicates a two-tailed *t*-test *p*-value of <0.05 compared to the untreated cohort. KS: keratan sulfate, TBG: thyroxine-binding globulin, CAG: a ubiquitous promoter including a cytomegalovirus enhancer element, a chicken β-actin promoter, and an intron., F: female, M: male, WT: wild-type.

**Figure 7 ijms-23-12693-f007:**
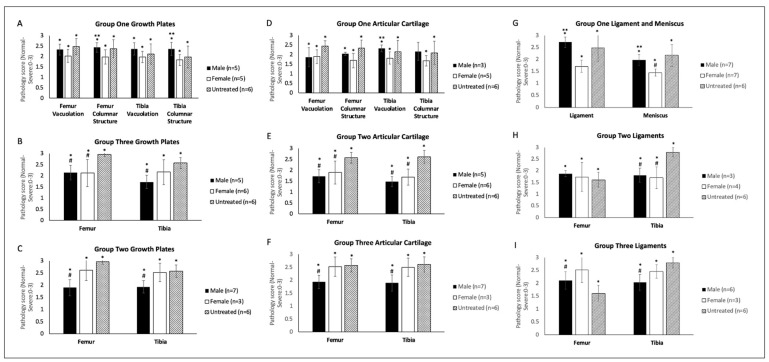
Histopathology severity ratings for (**A**) femoral and tibial growth plates of Group One (mice treated with the TBG promoter; 5 × 10^13^ GC/kg body weight), (**B**) femoral and tibial growth plates of Group Two (mice treated with the CAG promoter; 5 × 10^13^ GC/kg body weight), (**C**) femoral and tibial growth plates of Group Three (mice treated with the TBG promoter; 2 × 10^14^ GC/kg body weight), (**D**) femoral and tibial articular cartilage of Group One (mice treated with the TBG promoter; 5 × 10^13^ GC/kg body weight), (**E**) femoral and tibial articular cartilage of Group Two (mice treated with the CAG promoter; 5 × 10^13^ GC/kg body weight), (**F**) femoral and tibial articular cartilage of Group Three (mice treated with the TBG promoter; 2 × 10^14^ GC/kg body weight), (**G**) femoral ligament and meniscus of Group One (mice treated with the TBG promoter; 5 × 10^13^ GC/kg body weight), (**H**) femoral and tibial ligaments of Group Two (mice treated with the CAG promoter; 5 × 10^13^ GC/kg body weight), and (**I**) femoral and tibial ligaments of Group Three (mice treated with the TBG promoter; 2 × 10^14^ GC/kg body weight). All WT severity scores were zero. * indicates a two-tailed *t*-test *p*-value of <0.05 compared to the WT cohort. ** indicates a two-tailed *t*-test *p*-value of <0.05 compared to the opposite sex. # indicates a two-tailed *t*-test *p*-value of <0.05 compared to the untreated cohort. TBG: thyroxine-binding globulin, CAG: a ubiquitous promoter including a cytomegalovirus enhancer element, a chicken β-actin promoter, and an intron.

**Figure 8 ijms-23-12693-f008:**
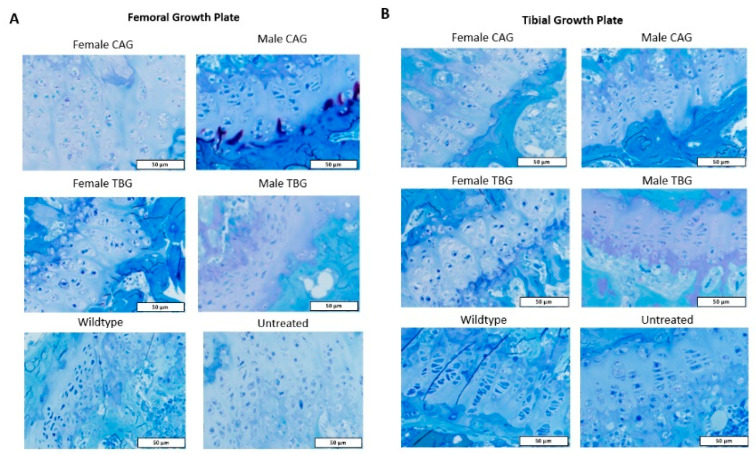
Toluidine blue staining of AAV8 CAG-hGALNS (Group Two) and TBG-hGALNS (Group Three) treated male and female (**A**) femoral growth plates, (**B**) tibial growth plates.

**Figure 9 ijms-23-12693-f009:**
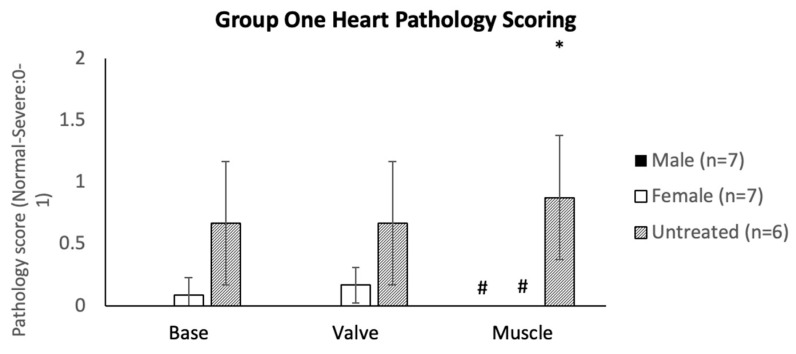
Heart pathology scoring of Group One (mice treated with the TBG promoter; 5 × 10^13^ GC/kg body weight) ranged from normal (0) to severe (1). Error bars display standard deviation. * indicates a two-tailed *t*-test *p*-value of <0.05 compared to the opposite sex. # indicates a two-tailed *t*-test *p*-value of <0.05 compared to the untreated cohort.

## Data Availability

Data will be available upon request.

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
