# Peer review of "Sex Difference Leads to Differential Gene Expression Patterns and Therapeutic Efficacy in Mucopolysaccharidosis IVA Murine Model Receiving AAV8 Gene Therapy"

_ijms, 2022, doi:10.3390/ijms232012693_

Round 1
Reviewer 1 Report
The manuscript "Sex difference leads to differential gene expression patterns 2 and therapeutic efficacy in Mucopolysaccharidosis IVA murine 3 model receiving AAV8 Gene Therapy" by Piechnik et al. is an interesting descriptive manuscript that explore further one of the challenges of gene therapy, the immune response against AAV vectors and the gender differences. There are reports in literature regarding the differences in neutralizing antibody titers in human not only by geographic region but also by gender as the major reasons for global differences (excluding of course the method, cutoff titer and other variables that difficult side by side comparison between studies).
This manuscript expands in this knowledge, without further exploring the mechanistic reasons for these findings. Maybe, exploration using an immunocompromised mice model will provide further insights in the reason of the gender differences.
I don't have major comments, the introduction is nicely written with a clear explanation of the problem and the current state of the art. The discussion could be expanded to bring some reports from human neutralizing antibody levels.
As a minor comments, I would suggest modifying the style of presentation of the results. The higher variability is some test makes the bar graphs the less useful of the charts to be used. Figure 2 can benefit of a 2D line with markers that allow a easy time-follow up. Even combining the 3 groups in same graph.
Similar comment for the next figures, were scatter plots can be more useful.
Figure 7 is really small to analyze properly all the data present in it.
Figure 9, SD higher cut is not visible.
Author Response
Comment: The discussion could be expanded to bring some reports from human neutralizing antibody levels.
Response: Thank you for your comment. We have included relevant sources to characterize the role of human neutralizing antibodies in the immune response to AAV-mediated gene therapy.
Comment: I would suggest modifying the style of presentation of the results. The higher variability is some test makes the bar graphs the less useful of the charts to be used. Figure 2 can benefit of a 2D line with markers that allow a easy time-follow up. Even combining the 3 groups in same graph. Similar comment for the next figures, were scatter plots can be more useful.
Response: We agree to change the figures from bar graphs to line graphs since time course is included. We changed Figures 2, 3, and 4 to chart-type linear 2D incorporate markers for each group to delineate change over time.
Comment: Figure 7 is really small to analyze properly all the data present in it. Figure 9, SD higher cut is not visible.
Response: The orientation of Figure 7 changed to horizontal for a better view of the results.
Comment: Figure 9, SD higher cut is not visible.
Response: The maximum values in the Y axis were increased to visualize the cut better.
Reviewer 2 Report
In this manuscript, Piechnik et al are exploring the response to an AAV8 based gene therapy for MPSIVa based on sex in a MPSIVa mouse knockout model. Overall, the manuscript is well-written, the data support their assertion that sex differences exist in the response to AAV8-GALNS, and the topic is of high interest to the gene therapy community. The manuscript is recommended for publication upon addressing the following concern:
- Although sex differences do exist in several of the individual assays presented in this manuscript, the overall results don’t overwhelmingly suggest that sex differences play a large role in the outcomes of the MPSIVa mice. For example, in figure 2, there are fairly large differences in plasma hGALNS activity between the groups of male and female knockout mice; however, in all cases the females groups still exhibited supraphysiological levels of GALNS activity compared the wt baseline levels. Additionally, some of the other sex differences, although significantly different, are quite modest, specifically for the pathology scoring in figures 7 and 9. It would be helpful to the readers if the authors would note these points and provide short discussions/explanations in their results or discussion sections.
- There is no description in the results section of the experiment shown in Figure 8. Please add.
- For the histopathological analyses, it would be helpful to provide a description or table of the scoring criteria in the methods section.
Author Response
Comment: Although sex differences do exist in several of the individual assays presented in this manuscript, the overall results don’t overwhelmingly suggest that sex differences play a large role in the outcomes of the MPSIVa mice. For example, in figure 2, there are fairly large differences in plasma hGALNS activity between the groups of male and female knockout mice; however, in all cases the females groups still exhibited supraphysiological levels of GALNS activity compared the wt baseline levels. Additionally, some of the other sex differences, although significantly different, are quite modest, specifically for the pathology scoring in figures 7 and 9. It would be helpful to the readers if the authors would note these points and provide short discussions/explanations in their results or discussion sections.
Response: Thank you for your comment. We have modified our manuscript to emphasize that, although significant differences exist in GALNS activity, they appear modest in scope. The effects of these differences are evident by modest pathological scoring differences, and we therefore suggest further studies to explore the clinical significance of these differences. The implications of these differences would be interesting to explore in further studies as they relate to overall disease prognosis. We elaborate these limitations of our study in our discussion section.
Comment: There is no description in the results section of the experiment shown in Figure 8. Please add.
Response: The description of the results has been added.
Comment: For the histopathological analyses, it would be helpful to provide a description or table of the scoring criteria in the methods section.
Response: Thank you for your comment. We added our method in materials and method section. Basically, we have evaluated for the reduction of storage by pathological scoring in a blind manner (Tomatsu et al. Mol Ther. 18(6), 1094-1102, 2010; Tomatsu et al. Mol Genet Metab. 114(2), 195-202, 2015).
We have revised the text as follows;
Assessment of reduction of storage materials were described previously [58,59]. Briefly, tissues from treated, untreated, and wild-type mice were evaluated for reduction in storage without knowledge of their treatment. The analysis was performed with three blind readers to evaluate the severity of a scalar model of normal to severe (0 to 3) for the femoral and tibial growth plate, articular cartilage, and ligaments. “No storage or very slight” was 0, “slight but obvious” was 1, “moderate” was 2, and “marked” was 3. In the first study, cardiac tissue pathology was evaluated similarly on a normal to severity scale of 0 to 1. Averages of individual severity scores were calculated and reported. Mann–Whitney U test was used to compare the score between untreated and treated mice.